# Changes in Endogenous Oxytocin Levels and the Effects of Exogenous Oxytocin Administration on Body Weight Changes and Food Intake in Polycystic Ovary Syndrome Model Rats

**DOI:** 10.3390/ijms23158207

**Published:** 2022-07-26

**Authors:** Shota Yamamoto, Hiroki Noguchi, Asuka Takeda, Ryosuke Arakaki, Maimi Uchishiba, Junki Imaizumi, Saki Minato, Shuhei Kamada, Tomohiro Kagawa, Atsuko Yoshida, Takako Kawakita, Yuri Yamamoto, Kanako Yoshida, Masafumi Kon, Nobuo Shinohara, Takeshi Iwasa

**Affiliations:** 1Department of Obstetrics and Gynecology, Institute of Biomedical Sciences, Graduate School, Tokushima University, Tokushima 770-8501, Japan; c202290001@tokushima-u.ac.jp (S.Y.); 2Department of Renal and Genitourinary Surgery, Graduate School of Medicine, Hokkaido University, Sapporo 060-0808, Japan

**Keywords:** PCOS, body weight, food intake, metabolic, oxytocin, DHT

## Abstract

Polycystic ovary syndrome (PCOS) is frequently seen in females of reproductive age and is associated with metabolic disorders that are exacerbated by obesity. Although body weight reduction programs via diet and lifestyle changes are recommended for modifying reproductive and metabolic phenotypes, the drop-out rate is high. Thus, an efficacious, safe, and continuable treatment method is needed. Recent studies have shown that oxytocin (OT) reduces body weight gain and food intake, and promotes lipolysis in some mammals, including humans (especially obese individuals), without any adverse effects. In the present study, we evaluated the changes in endogenous OT levels, and the effects of acute and chronic OT administration on body weight changes, food intake, and fat mass using novel dihydrotestosterone-induced PCOS model rats. We found that the serum OT level was lower in PCOS model rats than in control rats, whereas the hypothalamic OT mRNA expression level did not differ between them. Acute intraperitoneal administration of OT during the dark phase reduced the body weight gain and food intake in PCOS model rats, but these effects were not observed in control rats. In contrast, chronic administration of OT decreased the food intake in both the PCOS model rats and control rats. These findings indicate that OT may be a candidate medicine that is efficacious, safe, and continuable for treating obese PCOS patients.

## 1. Introduction

Polycystic ovary syndrome (PCOS) is an endocrine disorder that is frequently seen in females of reproductive age, and its estimated prevalence is 5% to 16% [1,2,3]. However, the etiology of PCOS remains unclear. It is well-known that PCOS is one of the causes of anovulation and subsequent infertility [4], and that PCOS may increase the risk of metabolic disorders such as obesity, insulin resistance, type 2 diabetic mellitus, and hyperlipidemia [5,6]. In addition, it is generally known that such reproductive and metabolic phenotypes of PCOS are exacerbated by excessive weight gain and obesity [5,6]. For that reason, physicians often recommend diet and lifestyle modifications for body weight reduction to obese PCOS patients [7,8]; however, the adherence rate is low, with as many as 50% to 70% of patients dropping out of the body weight reduction programs, and few PCOS patients succeed in reducing their body weight [7,9]. Furthermore, although some physicians prescribe diet pills and oral antidiabetic medicines for reducing body weight, their efficacy and safety remain unclear [7,8]. As such, the development of an efficacious, safe, and continuable method of body weight reduction is needed for PCOS patients.

Oxytocin (OT) is a 9-amino acid neuropeptide synthesized by the paraventricular nucleus and supraoptic nucleus in hypothalamic regions, which is secreted from the posterior lobe of the pituitary gland [10]. Recent studies have revealed that OT plays an important role in controlling the metabolism, appetite, and body weight of humans and animals [11]. Previous studies also found that the intraperitoneal, subcutaneous, intracerebroventricular, or intranasal injection of OT not only reduces food intake and/or body weight, but also activates lipolysis in adipose tissue brought on by β-oxidation, and reduces the fat weight in some mammals, such as mice [12,13,14,15,16,17,18], rats [18,19,20,21,22,23,24,25], monkeys [26], and humans [27,28]. In addition, these metabolic effects of OT are stronger in diet-induced and genetically obese rodents [14,29,30] and obese humans [28], and there have been no reported adverse effects of OT. Thus, OT may be useful for the treatment of PCOS patients. However, since research in humans is complicated, preliminary studies using animals are needed.

In our most recent study, we established novel PCOS model rats implanted with a silicon tube containing dihydrotestosterone (DHT) diluted with peanut oil [31]. These novel PCOS model rats have similar reproductive and metabolic features as PCOS patients, and unlike the conventional PCOS models, they do not show atrophic degeneration of the ovaries and uterus; as such, their features are more like human PCOS than those of the conventional models [31].

In this study, we examined the effects of the administration of OT as a therapeutic agent on the metabolic disorders in PCOS model rats. We also studied the changes in endogenous OT in the PCOS model rats.

## 2. Results

### 2.1. Reproductive and Metabolic Phenotypes

The body weight was higher in the PCOS group than in the Control group until the age of 84 days (two-way ANOVA: group, F(1,143) = 373.841, *p <* 0.001; Day, F(8,143) = 755.475, *p <* 0.001; interaction, F(8,144) = 12.413, *p <* 0.001; Figure 1A). Similarly, the body weight change was larger in the PCOS group (two-way ANOVA: group, F(1,143) = 435.250, *p <* 0.001; Day, F(8,143) = 727.508, *p <* 0.001; interaction, F(8,144) = 13.516, *p <* 0.001; Figure 1B), and the cumulative food intake was higher in the PCOS group (two-way ANOVA: group, F(1,143) = 113.649, *p <* 0.001; Day, F(8,143) = 1555.943, *p <* 0.001; interaction, F(8,144) = 6.161, *p <* 0.001; Figure 1C) than in the Control group. The feed efficiency was lower in the PCOS group than in the Control group (Student’s *t*-test: *p <* 0.01; df =14, *t =* 3.667; Figure 1D). In addition, the PCOS group had significantly more subcutaneous fat (Student’s *t*-test: *p <* 0.01; d = −3.80), visceral fat (Student’s *t*-test: *p <* 0.01; df = 14, *t =* −3.33), and total fat (Student’s *t*-test: *p <* 0.01; df = 14, *t =* −3.33) than the Control group (Figure 1E–G).

The Control group showed regular 4- to 5-day estrous cycles, whereas the PCOS group showed acyclic or irregular cycles (Figure 2A). Representative ovaries from the Control group showed normal morphologies, whereas those from the PCOS group showed a polycystic morphology (Figure 2B). The number of estrous stages was significantly smaller in the PCOS group than in the Control group (Student’s *t*-test: *p <* 0.01; df = 14, *t =* 4.830), and the PCOS group tended to have heavier ovaries and a lighter uterus, but not to a statistically significant degree (Figure 2C).

As mentioned above, the rats chronically administered DHT, i.e., the PCOS group, showed PCOS-like features, which is consistent with our previous study [31].

### 2.2. Central and Peripheral OT Levels

The serum OT level at 90 days of age was significantly lower in the PCOS group than in the Control group (Student’s *t*-test: *p <* 0.05; df = 14, *t =* 2.39), whereas the hypothalamic mRNA levels of OT, OTR, NPY, AgRP, POMC and pporexin, and the level of OTR in visceral fat did not differ between the PCOS and Control groups (Figure 3). Thus, the result for the serum OT level was discordant with the result for the hypothalamic OT mRNA level in the PCOS group.

### 2.3. Effects of Acute OT Administration on Food Intake and Body Weight

In the light phase, the body weight change and food intake after OT administration did not differ from those after saline administration in both the PCOS and Control groups (Figure 4A). In contrast, in the dark phase, the body weight change was significantly reduced by OT administration in the PCOS group, and it was not affected by OT administration in the Control group (Student’s *t*-test: *p <* 0.05; df = 14, *t =* 3.20); a similar result was seen for food intake in the PCOS group (Student’s *t*-test: *p <* 0.05; df = 14, *t =* 2.62; Figure 4B).

### 2.4. Effects of Chronic OT Administration on Food Intake and Body Weight

The body weight change did not differ significantly between the saline-injected and OT-injected periods in both the PCOS and Control groups. On the other hand, the cumulative food intake was significantly less during the OT-injected period than during the saline-injected period in the PCOS group (two-way ANOVA: group, F(1,111) = 11.565, *p <* 0.01; Day, F(6,111) = 247.756, *p <* 0.01; interaction, F(6,111) = 1.563, not significant) and in the Control group (two-way ANOVA: group, F(1,111) = 77.633, *p <* 0.01; Day, F(6,111) = 696.442, *p <* 0.01; interaction, F(6,111) = 6.176, *p <* 0.01; Figure 5).

## 3. Discussion

PCOS is a common disease with a poorly known etiology in females of reproductive age [1,2,3], and it increases the risk of reproductive and metabolic disorders [4,5,6]. In addition, it is known that these reproductive and metabolic disorders are exacerbated by obesity [5,6]. Even though physicians often recommend body weight reduction programs, such as dieting and exercise programs, for diet and lifestyle modifications to PCOS patients, the drop-out rate from such programs is high [7,9]. For these reasons, a continuable therapeutic approach to body weight reduction is needed for PCOS patients.

It is becoming increasingly obvious that OT, a neuropeptide synthesized in the hypothalamus, is involved in controlling metabolism, appetite, and body weight [11]. In our previous study, we found that the body weight and food intake were reduced by chronic peripheral OT administration in conventional PCOS model rats [32]. Thereafter, we established novel PCOS model rats showing reproductive and metabolic features that are more similar than those of conventional PCOS model rats to the features of human PCOS [31]. The present study is the first to evaluate the changes in endogenous OT levels and the effects of OT administration on body weight reduction in the novel PCOS model rats. In this study, we found that the serum OT level in rats was significantly lower in the PCOS group than in the Control group, indicating that the endogenous OT level might be decreased in human PCOS patients. In previous studies, it has been determined that the endogenous OT level is decreased in fatty rats and obese humans [16,33,34,35], and it is considered that the same pathophysiological mechanisms are involved in obese PCOS individuals. However, in the present study, we found no significant differences in the hypothalamic mRNA levels of OT, OTR, NPY, AgRP, POMC, and pporexin between the PCOS and Control groups. It is unclear why this discrepancy between the hypothalamic OT mRNA level and the serum OT level was seen in the PCOS group. One possibility is that the production of OT at the central level was not increased, whereas its secretion to blood might be decreased or/and OT consumption in peripheral tissues might be high in the PCOS group by some unclear mechanisms. Another possibility is that the mRNA level of hypothalamic OT does not reflect the protein level in the PCOS group. Further examinations are needed to clarify the precise mechanisms.

In this study, acute peripheral OT administration did not affect the body weight or food intake of the PCOS group during the light phase, but it reduced the body weight and food intake of the PCOS group during the dark phase. It was considered that since rats are nocturnal animals that are active and increase their food intake during the dark phase [36], OT administration just before the dark phase, i.e., just before the food intake increase, is effective for reducing the body weight and food intake. In addition, these effects of OT on body weight and food intake were not observed in the Control group in this study. As noted above, it has been reported that the effects of exogenous OT are more evident in obese individuals in both humans [28] and experimental animals, such as rodents [14,29,30] and monkeys [26]. The results from our acute administration study indicated the possibility that OT administration would be more effective in obese PCOS patients, and that chronic administration may reduce their body weight and improve the metabolic and reproductive phenotypes. Thus, the effects of chronic OT administration were also examined. We found that chronic peripheral OT administration reduced the food intake in both the PCOS and Control groups, but it did not significantly affect the body weight in these groups. We speculated that long-term (seven days) OT administration may be more effective than acute (single) administration of OT since it affected not only the PCOS group, but also the Control group. In addition, it is possible that the effects of OT administration on body weight during the dark phase may be cancelled out by weight gain during the light phase. If so, it is likely that OT administration in longer periods or during both the dark and light phases would reduce the body weight.

As mentioned above, these results suggest the possible beneficial effects of OT administration for the treatment of PCOS. In particular, it is possible that the effects of exogenous OT might be more evident in obese PCOS patients whose level of endogenous OT is attenuated. In addition, because it has been shown that long-term OT administration does not cause any obvious adverse effects, such as liver and kidney injury, febrile responses, and behavioral changes [37,38], OT is expected to be an effective and safe treatment for PCOS. In the future, further examinations are needed to clarify whether the same effects can be observed in human PCOS patients.

## 4. Materials and Methods

### 4.1. Animals

For this study, 22-day-old female Wistar rats were purchased from Charles River Laboratories Japan (Yokohama City, Kanagawa, Japan). They were housed in a room with a controlled temperature (24 °C) and 12-h light/dark cycle (lights were turned on at 08:00 and turned off at 20:00), and had free access to standard food (type MF; Orient Yeast Co., Ltd., Tokyo, Japan) and water. Surgical procedures and tissue sampling, which were carried out under sevoflurane-induced anesthesia, were performed according to the ethical standards of the Animal Care and Use Committee of Tokushima University.

At 28 days of age, the female Wistar rats were divided into a PCOS model rat group (PCOS group, *n =* 8) and a control rat group (Control group, *n =* 8). Based on our previous study, rats in the PCOS group were implanted with a silicon tube (As One Co., Ltd., Tokyo, Japan; inner diameter, 3 mm; outer diameter, 5 mm; length of the filled part, 10 mm) containing diluted DHT. The DHT was dissolved in a solution of 80% peanut oil and 20% ethanol to a concentration of 16 mg/mL. In the Control group, each rat was implanted with an empty tube to exclude the effect of surgical invasion. After the surgery, all rats were individually housed.

### 4.2. Analysis of the Reproductive and Metabolic Phenotypes and Tissue Sampling

The body weight and food intake were measured every week starting from the surgical day (at 28 days of age) and 1 week after the surgical day (at 35 days of age), respectively. In addition, starting 3 weeks from the surgical day (at 49 days of age), daily vaginal epithelial smears were collected to evaluate the estrous cyclicity for 9 days. To collect smears, a glass pipette filled with sterilized water was inserted into the vaginal orifice to a depth of 5 mm, and the vagina was flashed several times. The collected fluid sample was dropped onto a slide glass and air-dried. Then, the slide glass was stained with Giemsa stain to observe the types of cells on the slide, and cytological examination of the estrous cycle (proestrus, estrus, metestrus, and diestrus) was performed.

At 90 days of age, all rats were killed by decapitation under sevoflurane-induced anesthesia, and the blood, brain, ovaries, and visceral fat were harvested. At that time, the weights of the visceral fat (the parametrial, perirenal, and mesenteric deposits), subcutaneous fat (the inguinal deposit), bilateral ovaries, and uterus were measured. Visceral fat and subcutaneous fat were weighed immediately after being excised, and a small sample of visceral fat tissue was collected. Whole blood was centrifuged, and the serum was collected and stored at −20 °C. The left ovaries were fixed in 4% paraformaldehyde. Frozen brain and fat samples were used for the peripheral and central mRNA assays, serum samples were used for measuring the concentrations of OT, and the fixed ovaries were used for histological analyses.

### 4.3. Evaluation of the Effects of Acute OT Administration on Food Intake and Body Weight

In addition to the evaluation of the endogenous OT levels in the PCOS model rats, the effects of acute exogenous OT administration on food intake and body weight were also evaluated in the rats. At 70 days of age, the basal levels of body weight changes and food intake during the light phase (08:00 to 20:00) and dark phase (20:00 to 08:00) were measured over 3 consecutive days (the saline-injected period) in both groups. Afterwards, both groups received acute intraperitoneal administrations of OT (1600 μg/kg, 0.4 to 0.5 mL injection volume) just before the initiation of the dark phase (20:00) and the light phase (08:00; the OT-injected period), and the food intake and body weight were measured during these two phases. The dose of OT was determined with reference to a previous study [12]. The mean body weight change and food intake were compared between the saline-injected and OT-injected periods.

### 4.4. Evaluation of the Effect of Chronic OT Administration on Food Intake and Body Weight

Another cohort of 28-day-old Wistar rats was divided into a PCOS group (*n =* 8) and a Control group (*n =* 8) to evaluate the effects of chronic OT administration on the body weight and food intake. At 10 weeks after the surgical day, all rats were intraperitoneally injected with saline for 7 consecutive days, then injected with OT (1200 μg/kg, 0.4 to 0.5 mL injection volume) for the following 7 consecutive days. The dose of OT was determined with reference to our previous study [37].

### 4.5. Hormone Assay

Whole blood was centrifuged at 3000 rpm for 20 min at 4 °C, and the serum was sent to a commercial laboratory (ASKA Pharmaceutical Medical Inc., Co., Ltd., Fujisawa City, Kanagawa, Japan) for the measurement of the OT levels by a chemiluminescent enzyme immunoassay. In this assay, serum was diluted with a buffer mixture containing bovine serum albumin and 0.1% trifluoroacetic acid, and the elution was evaporated and reconstructed with a solid phase extraction column. Afterwards, the samples were added to goat anti-rabbit immunoglobulin G antibody pre-coated wells with alkaline phosphatase conjugated-OT and rabbit antibody specific to OT. After washing, a chemiluminescent assay for alkaline phosphatase was performed. The limit of detection for serum OT was 15 pg/mL, and as this assay does not detect Arg-vasopressin, the results were not affected by vasopressin.

### 4.6. Histology

Representative ovaries were embedded in paraffin and sliced into 4-μm-thick sections, then the sections were stained with hematoxylin and eosin. The Zeiss Imager M2 microscope and AxioVision (version 4.8) acquisition software (Zeiss, Oberkochen, Germany) were used to capture histological images.

### 4.7. Real-Time Polymerase Chain Reaction (PCR)

Whole hypothalamic explants were dissected from the frozen brains, and the mRNA expression levels were assayed. The brain sections were dissected out via an anterior coronal cut at the anterior border of the optic chiasm, a posterior cut at the posterior border of the mammillary bodies, parasagittal cuts along the hypothalamic fissures, and a dorsal cut 2.5 mm from the ventral surface. Total mRNA was isolated using a TRIzol reagent kit (Invitrogen Co., Carlsbad, CA, USA) and an RNeasy^®^ mini kit (Qiagen GmbH, Hilden, Germany). cDNA was synthesized using the SuperScript III First-Strand Synthesis System for Real-Time PCR (Invitrogen Co., Life Technologies Japan Ltd., Minato Ward, Tokyo, Japan), the StepOnePlus^TM^ Real-Time PCR System (PE Applied Biosystems, Foster City, CA, USA), and Fast SYBR^®^ green. Then, the mRNA expression levels of OT, OT receptor (OTR), neuropeptide Y (NPY), agouti-related protein (AgRP), pro-opiomelanocortin (POMC), and prepro-orexin (pporexin) were quantified. The mRNA expression levels of OT in visceral fat were also measured. The mRNA expression levels were normalized to that of the *GAPDH* or *18S rRNA* housekeeping gene. Dissociation curve analysis was also performed for each gene at the end of the PCR. Each amplicon generated a single peak. The primer sequences, product sizes, and annealing temperatures are shown in Table 1. The PCR conditions were as follows: initial denaturation and enzyme activation at 95 °C for 20 s, followed by 45 cycles of denaturation at 95 °C for 3 s, and annealing and extension for 30 s.

### 4.8. Statistical Analyses

All results are expressed as the means ± standard error of the mean (SEM). The student’s *t*-test for parametric data or the Mann–Whitney U test for non-parametric data was used for statistical investigations between the PCOS and Control groups or the saline-injected and OT-injected periods. Two-way repeated-measures analysis of variance (ANOVA) was used for comparisons of the overall changes in body weight and food intake between the PCOS and Control groups. *p*-values of <0.05 were considered to be statistically significant.

## 5. Conclusions

This is the first study to evaluate the changes in endogenous OT in PCOS model rats, and we found that the serum OT level was decreased in the PCOS model rats. Acute administration of OT during the dark phase reduced the body weight gain and food intake only in PCOS model rats, and seven days of OT administration decreased the food intake in both the PCOS model rats and control rats. We previously reported that the administration of OT does not cause any obvious adverse effects, such as liver and kidney injury, febrile responses, and behavioral changes. Taken together, our results indicate that OT may be useful as an efficacious, safe, and continuable medicine for treating obese PCOS individuals.

## Figures and Tables

**Figure 1 ijms-23-08207-f001:**
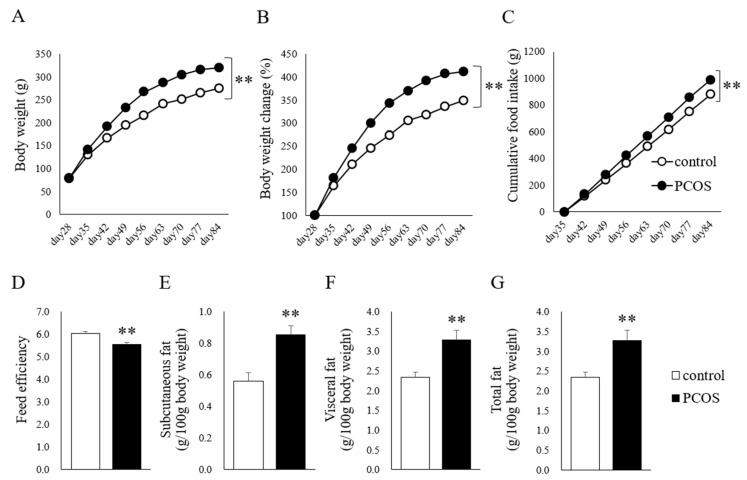
(**A**) Body weight, (**B**) body weight change (% initial body weight), (**C**) cumulative food intake, (**D**) feed efficiency, (**E**) subcutaneous fat weight (g/100 g body weight), (**F**) visceral fat weight (g/100 g body weight), and (**G**) total fat weight (g/100 g body weight) in the PCOS (■) and Control (□) groups. Data are expressed as the means ± SEM. ** *p <* 0.01.

**Figure 2 ijms-23-08207-f002:**
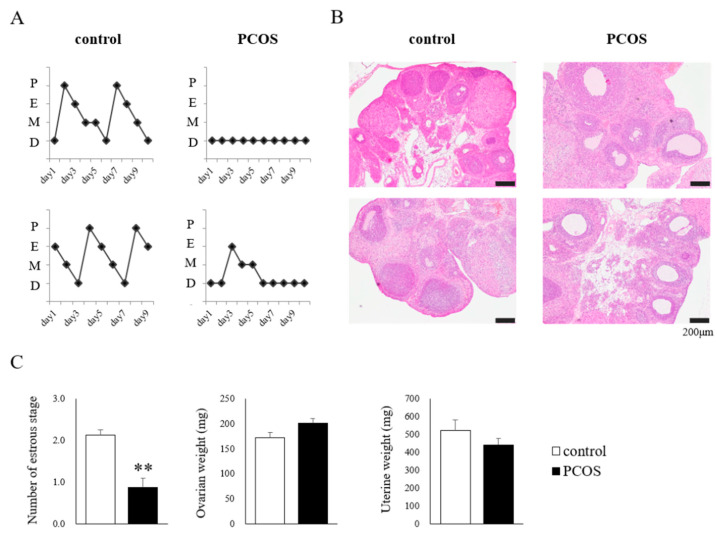
(**A**) The estrous cycle of two representative rats: D, diestrus; M, metestrus; E, estrus; P, proestrus. (**B**) Representative ovarian morphology. (**C**) The number of estrous stages, ovarian weight, and uterine weight in the PCOS (■) and Control (□) groups. Data are expressed as the means ± SEM. ** *p* < 0.01.

**Figure 3 ijms-23-08207-f003:**
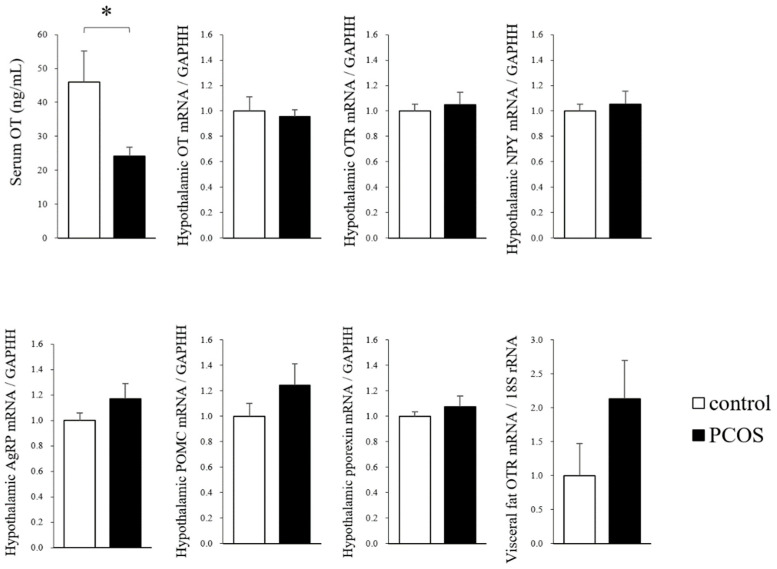
The serum oxytocin (OT) levels, the hypothalamic mRNA expression levels of OT, oxytocin receptor (OTR), neuropeptide Y (NPY), agouti-related protein (AgRP), pro-opiomelanocortin (POMC) and prepro-orexin (pporexin), and the visceral mRNA levels of OTR in the PCOS and Control groups. Data are expressed as the means ± SEM. * *p <* 0.05.

**Figure 4 ijms-23-08207-f004:**
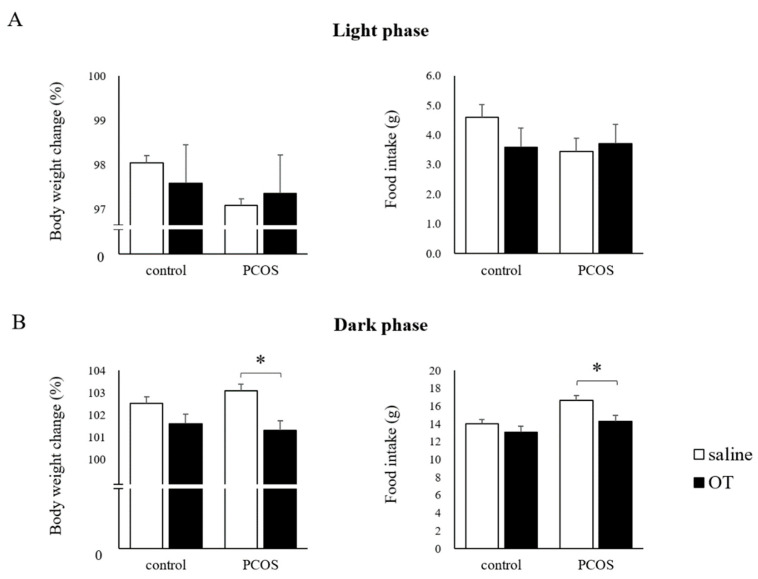
The body weight change and food intake after acute saline (□) or oxytocin (OT; ■) administration during the light phase (**A**) or the dark phase (**B**) in the PCOS and Control groups. Data are expressed as the means ± SEM. * *p* < 0.05.

**Figure 5 ijms-23-08207-f005:**
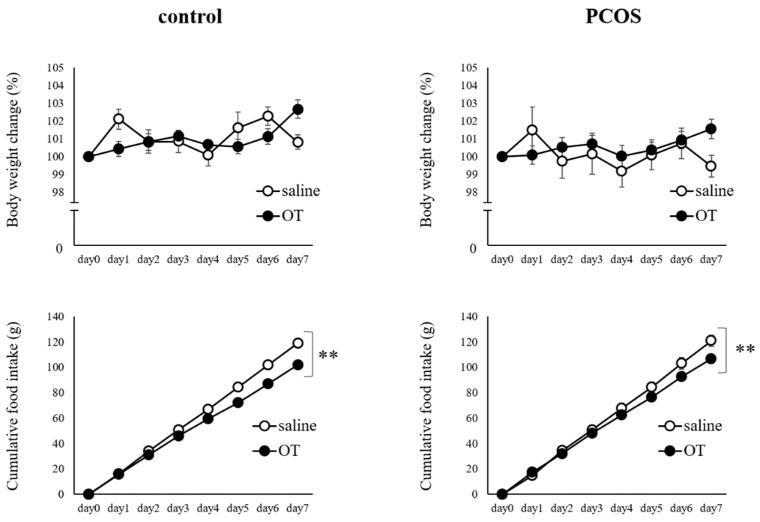
Body weight change (% initial body weight) and cumulative food intake during the 7 days of saline (○) or oxytocin (●) administration in the PCOS and Control groups. Saline was intraperitoneally injected in the saline groups, and oxytocin (1200 μg/kg) was intraperitoneally injected in the oxytocin groups at 2 h before the dark phase. Data are expressed as the means ± SEM. ** *p* < 0.01.

**Table 1 ijms-23-08207-t001:** Primer sequences, product sizes, and annealing temperatures.

Primer	Sequence	Annealing T (°C)
OT forward	GAA CAC CAA CGC CAT GGC CTG CCC	62
OT reverse	TCG GTG CGG CAG CCA TCC GGG CTA	
OTR forward	CGA TTG CTG GGC GGT CTT	67
OTR reverse	CCG CCG CTG CCG TCT TGA	
NPY forward	GGG GCT GTG TGG ACT GAC CCT	66
NPY reverse	GAT GTA GTG TCG CAG AGC GGA G	
AgRP forward	TGA AGA AGA CAG CAG CAG ACC	63
AgRP reverse	AAG GTA CCT GTT GTC CCAAGC	
POMC forward	CCT CAC CAC GGA AAG CA	66
POMC reverse	TCA AGG GCT GTT CAT CTC C	
pporexin forward	GCC GTC TCT ACG AAC TGT TG	60
pporexin reverse	CGA GGA GAG GGG AAA GTT AG	
GAPDH forward	ATG GCA CAG TCAAGG CTG AGA	70
GAPDH reverse	CGC TCC TG GAA GAT GGT GAT	
18S rRNA forward	GAC GGA CCA GAG CGA AAG C	64
18S rRNA reverse	AAC CTC CGA CTT TCG TTC TTG A	

## Data Availability

Not applicable.

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
