# Peer review of "Changes in Endogenous Oxytocin Levels and the Effects of Exogenous Oxytocin Administration on Body Weight Changes and Food Intake in Polycystic Ovary Syndrome Model Rats"

_ijms, 2022, doi:10.3390/ijms23158207_

Round 1
Reviewer 1 Report
The Authors present an interesting study on the possible involvement of oxytocin (Oxt) in PCOS on a rat model. They have measured serum levels of Oxt in dihydrotestosterone-induced PCOS rats, levels of hypothalamic mRNAs for Oxt and related peptides, and evaluated the effects of acute and chronic administration of exogenous Oxt on food intake and obesity parameters.
While this paper opens a novel path in the study and possible treatments of PCOS in humans, it should be improved before being suitable for publication.
The experimental plan should be clearly explained to the reader before the Results section. A bulleted list at the end on the Introduction section or a graphic representation at the beginning of the Results section would be appropriate to indicate the length of the DHT conditioning period and the times of various experimental steps. This would be particularly important in regards to the acute administration of Oxt, which does not appear clear because reported too shortly in the Results section, as light phase and dark phase.
A critical observation of this research is the lack of decrease in hypothalamic Oxt (and related peptides) mRNAs as compared with that observed in serum Oxt levels of the rats after 10 weeks DHT treatment. Although somehow discussed, this results, which probably depends on complex regulatory mechanisms of secretion/uptake, would require additional hypotheses by further literature search and analysis.
Finally, the English language needs, here and there, some revising.
Reviewer 2 Report
1. The study entitled "Changes in endogenous oxytocin levels and the effects of exogenous oxytocin administration on body weight changes and food intake in polycystic ovary syndrome model rats" is aimed at addressing the impact of oxytocin (OT) in PCOS rat models using novel DHT induced PCOS model system. The authors are requested to refer the DHT PCOS model system by Krishnan et al., 2020
2. In general, the objective and the model system used are appropriate.
3. A consistent decline in circulating OT indicates a depletion of OT after sustained stimulation with DHT.
4. An increase in subcutaneous, visceral and total fat in DHT induced group are associated with increased body weight.
5. In figure 4B, verify the statistical significance.
6. In figure 4B, the body weight changes graph, the Y axis should start with origin '0".
7. In figure 5, for body weight change, the Y axis should start with "0".
8. In figure 1B, for change in body weight, why the authors preferred Y axis from 100? Justify.
9. In figure 3, the Y axis may be changed to fold change (2DDct) in the expression of hypothalamic OT mRNA, in relation with the house keeping gene.
10. What is the rationale of assessing the body weight changes in light and dark phases? It is strongly recommended to measure the levels of melatonin and correlate the changes to melatonin levels.
